# Hyaluronan in the Cancer Cells Microenvironment

**DOI:** 10.3390/cancers15030798

**Published:** 2023-01-28

**Authors:** Evgenia Karousou, Arianna Parnigoni, Paola Moretto, Alberto Passi, Manuela Viola, Davide Vigetti

**Affiliations:** Department of Medicine and Surgery, University of Insubria, 21100 Varese, Italy

**Keywords:** hyaluronan, soluble factors, ECM-extracellular matrix

## Abstract

**Simple Summary:**

The extracellular matrix has a complex structure, and the glycans within are difficult to study, but are able to elicit multiple effects on resident and immune cells, with a crucial role being exerted by hyaluronan.

**Abstract:**

The presence of the glycosaminoglycan hyaluronan in the extracellular matrix of tissues is the result of the cooperative synthesis of several resident cells, that is, macrophages and tumor and stromal cells. Any change in hyaluronan concentration or dimension leads to a modification in stiffness and cellular response through receptors on the plasma membrane. Hyaluronan has an effect on all cancer cell behaviors, such as evasion of apoptosis, limitless replicative potential, sustained angiogenesis, and metastasis. It is noteworthy that hyaluronan metabolism can be dramatically altered by growth factors and matrikines during inflammation, as well as by the metabolic homeostasis of cells. The regulation of HA deposition and its dimensions are pivotal for tumor progression and cancer patient prognosis. Nevertheless, because of all the factors involved, modulating hyaluronan metabolism could be tough. Several commercial drugs have already been described as potential or effective modulators; however, deeper investigations are needed to study their possible side effects. Moreover, other matrix molecules could be identified and targeted as upstream regulators of synthetic or degrading enzymes. Finally, co-cultures of cancer, fibroblasts, and immune cells could reveal potential new targets among secreted factors.

## 1. Introduction

Cancer is the result of the uncontrolled proliferation of cells that have usually undergone genetic mutations due to hereditary, environmental, and lifestyle factors. The pathological progression is characterized by persistent inflammation of the surrounding tissue, with a drastic change in the extracellular matrix (ECM) architecture. Indeed, multiple inflammatory signals produced by cancer cells and innate immune cells recruited in the tumor niche induce changes in the stroma that, in turn, influence the homeostasis of ECM, creating a “cancerized” microenvironment that supports tumor growth and metastasis [1]. The pathogenesis of cancer is characterized by several constant cell behaviors, such as self-sufficiency in growth signals, insensitivity to anti-growth signals, evasion of apoptosis, limitless replicative potential, activation of angiogenesis, and metastasis. These features can be sustained and/or affected by the ECM proteins (the so-called matrisome), and indeed among all the molecules, by the polysaccharide hyaluronan (HA) [2,3]. The ECM is an intricate network of protein fibers (collagen and elastin) held together by carbohydrate strings (proteoglycans, HA, and glycoproteins). Glycosaminoglycans (GAGs) are polysaccharide chains that constitute the glucidic moieties of proteoglycans. GAGs can be grouped into four subfamilies named chondroitin/dermatan sulfate (CS/DS), heparan sulfate/heparin (HS/HE), HA and keratan sulfate (KS), depending on the chain composition in specific disaccharide repeats in their glycosidic bonds and protein linkages, as well as sulfation pattern and degree. Almost all the GAG chains are covalently linked to a protein core, thus contributing to the proteoglycan native folding and functions. The only exception is HA, which is not bound to any polypeptide chain. Moreover, among all GAGs, only HA is synthesized at the plasma membrane, while the others are polymerized within the Golgi.

The increase in HA in the ECM of tumors can be attributed to the combined metabolism of cancerous and stromal cells, i.e., fibroblasts, endothelial cells (ECs), immune cells, and stem cells. Several molecules of the ECM undergo alterations in their concentration or structural/chemical modification due to cancer progression, and it is known that HA surrounds the tumor mass, is present in the stroma, and/or encircles cancerous cells [4].

The concept that HA could influence invasive tumor growth dates back to the end of the seventies when Bryan P. Toole and collaborators proposed that neoplastic cells needed the same HA-rich ECM of migrating cells, for example during embryonic development to invade local host tissues [5]. In the next few years, pioneering studies demonstrated that HA production by stromal cells is stimulated by interactions with tumor cells, and that HA synthesis is also increased in malignant cancer cells themselves [6,7,8]. Further, it was evident in many cancers, such as breast and ovarian carcinomas, that the degree of HA accumulation strongly correlates with the advancement of the malignancy and unfavorable prognosis of the patients [9,10,11]. It became clear that HA influences cancer progression modulating invasive vs. non-invasive status of neoplastic cells and favoring the formation of highly “malleable” tissues, with drastic changes in ECM architecture, density, and permeability, triggering pro-tumorigenic intracellular signaling cascades [11,12]. Nowadays, it is well-accepted that HA contributes to all the Hanahan and Weinberg’s hallmarks of cancer and therefore it is not surprising that HA has become a very promising target in cancer research [2,13].

This review aims to summarize the various alterations of HA metabolism in cancers and their possible use as therapeutic targets. Recently, the uncharacterized secreted protein C10orf118, aka coiled-coil domain-containing protein 186 or Q7z3E2, was identified as a novel factor that activates crosstalk between cancer and stromal cells, particularly fibroblasts, causing HA enrichment of the cancer niche [14]. ECM-derived molecules such as endorepellin can influence the synthases responsible for high-molecular-mass HA (HMWHA) (i.e., hyaluronan synthases, HASes). Hymecromone (i.e., 4-methylumbelliferone, 4-MU), already largely used in in vitro experiments, acts as an in vivo HA inhibitor in diseases characterized by high HA deposition, such as cancers, but with changing fortunes [15,16,17]. Finally, antioxidants can block chemical HA fragmentation into oligos, which are generally involved in busting specific cancerous behaviors, above all neoangiogenesis and proliferation.

## 2. HA in Cancers: Proliferation and Neoangiogenesis

HA accumulation in the stroma of several tumors usually correlates with a poor prognosis. The effect of HA on cancer development depends on the ability of each target cell to respond to the molecule via specific surface receptors. In particular, the engagement of one or more of the HA-binding proteins, among which CD44, receptor for hyaluronan-mediated motility (RHAMM), lymphatic vessel endothelial hyaluronan receptor 1 (LYVE-1), HA receptor for endocytosis (HARE), and toll-like receptor 4 (TLR4), accounts for the perturbation of tissue homeostasis, eventually unbalancing cell motility, proliferation, apoptosis, and tissue remodeling [2,15] (Figure 1). Briefly, the most important events are related to HA/CD44 binding, which elicits the anti-apoptosis and chemoresistance of breast tumor cells via a mechanism involving protein kinase C (PKC) and the production of miR-21 [18]; RHAMM signals to kinases, calmodulin, and cytoskeletal proteins, which primarily modify adhesion and cell motility in response to HA binding [19]; HARE binds to HA as well as to other GAGs, triggering extracellular signal-regulated kinases 1/2 (ERK1/2) and NF-kB activation, with an important role in the formation of lymph node metastasis in prostate cancer [20,21,22,23]. Notably, TLR4, a member of a receptor family involved in multiple signaling pathways in cancers and immune cells, is a promising immunotherapeutic target. Indeed, its engagement transforms the interleukin (ILs) secretory sets of myeloid-derived suppressor cells into permissive and tumor-promoting multiple signals [24], thus connecting the HA-rich environment with the inflammatory condition. A summary of these mechanisms is depicted in Figure 1.

During cancer mass development, hypoxia is one of the main events promoting cellular adaptations. The adaptive responses to hypoxia are due to the stabilization and activation of specific hypoxia-inducible factors (HIFs), especially HIF-1α. Importantly, the expression of HIFs in ECs, which control vascular endothelial growth factor 165 (VEGF-165) release [25], can lead to the initiation of neoangiogenesis. The mechanism for the promotion or inhibition of ECs proliferation (dependent on the activation of the link proteins ezrin and merlin) also requires HA-derived oligos (oligoHA). In fact, HMWHA cannot induce endothelial sprouting by itself [26], but needs to be fragmented either enzymatically or chemically. As reported by Mo et al., after interacting with CD44, oligoHA trigger ezrin pathway activation, leading to endothelial proliferation; whereas, longer HA fragments activate merlin, thus inhibiting cell proliferation [27]. In particular, in low-density cell culture, CD44 ensures the aggregation of cytoskeletal proteins that maintain the phosphorylation of merlin and, therefore, cell proliferation [28]. Instead, the engagement of CD44 with HA mimics cell–cell contacts and induces cell growth arrest. Bono et al. also suggested that HA affects cell proliferation through the membrane protein laylin, which is critical for the assembly of tight junctions in epithelial cells [28]. Notably, in an artificial membrane system, Wolny et al. highlighted that CD44 interaction with larger HA polymers is essentially irreversible, while smaller HA species (molecular weight of ≤10 kDa) bind in a reversible manner to the receptor [29].

In pathological conditions characterized by chronic inflammation, such as cancers, the tissues surrounding blood vessels can be submerged by cytokines providing multiple pieces of information. In this context, HA can be produced by ECs after the induction of HAS2. The same stimuli change the HS chains, altering the vascular layer permeability [30]. Moreover, as reported by in vitro studies, the cell surface HS interacts with VEGF-165, enhancing the phosphorylation of VEGF receptor 2 (VEGFR-2), thus increasing mitogenic activity, as well as endothelial tube formation, i.e., inducing neoangiogenesis [26]. Altering HS on the cell membrane can, therefore, alter the effect of VEGF on the cells [30].

It is noteworthy that some of these same inflammatory cytokines, such as IL-1β and tumor necrosis factor-alpha (TNF-α), can also increase the expression of HAS3 in ECs (HAS1 is not expressed) without the release of HA in the stroma. Since HA can also be secreted as a pericellular coat by the cells, this scenario requires further investigation. In fact, apart from the differences in the kinetics and regulation, HAS2 and HAS3 also differ in the size of HA synthesized: HAS2 produces polysaccharides larger than 2 MDa; whereas, HAS3 synthesizes molecules ranging from 0.2 to 2.0 MDa [31], and it might as well be responsible for a different localization of the GAG in the tumor mass. HA of different molecular mass is known to regulate cell proliferation and motility in vivo [29]. HAS3 is highly expressed in several types of cancers, such as renal carcinoma [32] and oral cancer [33]; however, its role in cancer HA metabolism remains unclear, even if there are intriguing data about its presence in cancer extracellular vesicles, and therefore clues about its involvement in cell–cell exchange of information [34].

Controlling excessive HA production under pathological conditions is pivotal to reducing neoangiogenesis and limiting tumor spreading. A specific matrikine, endorepellin, an anti-angiogenic effector derived from the HSPG perlecan (highly enriched in various human tumors [35]) induces selective autophagy of HAS2 in ECs [36], thus decreasing the deposition of HA. A study focused on the effects of endorepellin revealed a new function for HAS2, which, as “an integral membrane protein shuttling to and from the plasma membrane, could play a role in forming the pre-autophagosomal structure, via complexing with ATG9A upon autophagic activation” [36]. In particular, endostatin treatment ameliorates the progression-free survival and overall survival in patients with advanced non-small cell lung cancer [37]. In this case, HAS3 is not affected, again underlying the different roles of the isoenzymes and the lack of knowledge of HAS3 functions.

## 3. Controlling HASes to Control HA

The presence of HA in tissues depends on the balance between its synthesis and degradation. The synthesis of HA requires HASes (HAS1-3), which are responsible for its synthesis and extrusion through the plasma membrane, and two cytosolic UDP-sugar substrates (i.e., UDP-D-glucuronic acid, UDP-GlcUA, and UDP-N-acetylglucosamine, UDP-GlcNAc) derived from energetic metabolism [38]. UDP-GlcUA production is catalyzed by UDP-glucose 6-dehydrogenase (UGDH), which uses NAD + coenzyme to oxidize UDP-glucose (Figure 2). Thus, UDP-GlcUA synthesis is affected by the cell redox potential. Furthermore, HASes compete for UDP-GlcUA as it is also used for detoxification (i.e., glucuronidation reactions) or for the synthesis of UDP-xylose, which is employed to attach GAGs to the core protein of proteoglycans (Figure 2) [39].

A common inhibitor of HA synthesis is 4-MU, a derivative of coumarin with the IUPAC name 7-hydroxy-4-methylcoumarin, and the international free name Hymecromone. This molecule has been widely used as a choleretic and an antispasmodic drug. However, 4-MU has been shown to inhibit HA production in multiple cell lines and tissue types, both in vitro and in vivo in at least two ways: acting as a competitive substrate for HASes, and reducing the expression of HAS2 mRNA [40] (Figure 1 and Figure 2). This modified coumarin is rapidly glucuronidated by UDP-glucuronosyltransferase (UGTs) enzymes, which exploit UDP-GlcUA as the donor substrate, eventually reducing the activity of HASes. The synthesis of the other glycoconjugates is only marginally affected by 4-MU [41]; for example, the synthesis of other GAGs occurs inside the Golgi, which is supplied by UDP-sugars by high-efficiency transporters that maintain a saturating concentration of precursors inside the lumen of this organelle [42]. In this light, 4-MU has been used to test the inhibition of tumor growth and diffusion in several animal cancer models, and indeed 4-MU is widely used in in vitro and in vivo HA studies [43,44,45,46,47,48]. The toxicological, pharmacokinetic, and pharmacodynamic aspects of the treatment regimen, as well as the biological effects of its metabolites and bioavailability, require further investigation to better understand 4-MU metabolism. In a recent study, 4-MU safety and effects were evaluated in mice: after 6 months of treatment, histopathological examination revealed mild atrophy in articular cartilage, non-affecting motor performance, and reduced perineuronal nets formation around neurons, enhancing memory retention. Incidentally, these results suggest that 4-MU treatment might offer a strategy for perineuronal net modulation in memory enhancement [49] without major side effects. In a mouse model of atherosclerotic lesions, the long-term use of 4-MU had a positive effect in reducing HA deposition but caused an acceleration of the disease due to severe damage to the endothelial glycocalyx [17]. Recently, patients with severe COVID-19 have displayed increased plasma HA levels, among other signs. In vitro, Human Identical Sequences of SARS-CoV-2 activated the expression of both adjacent and distant genes associated with inflammation. HAS2 was found to be one of the activated genes, eventually increasing HA accumulation. Considering that 4-MU inhibits HA production in SARS-transfected cells in vitro, it should be further investigated as a therapeutic option for preventing severe outcomes in COVID-19 patients [50].

Products containing 4-MU are available in the USA and Europe as inflammatory dietary supplements, but not yet for cancer therapies [15].

4-MU has also been used in several cancer cell lines to evaluate the role of HA in cell behaviors, with different results. In a model of human acute leukemia, 4-MU seems to exert an HA-independent suppressive effect [51]; in glioblastoma cells, 4-MU decreased HA synthesis, diminished proliferation, and induced apoptosis, while reducing cell migration and the activity of metalloproteinases [52]. In this cancer model, the absence of endogenous HA increased cell sensitivity to anticancer drugs, underlining its protective role in cells. In a model of liver fibrosis, the absence of HA caused by 4-MU stopped the hepatic stellate cells (HSCs) transdifferentiation to myofibroblasts, blocking the cancer progression of the liver [53].

The synthesis of HA is strictly related to the bioavailability of sugars, and therefore, UDP-glucose and UDP-GlcUA (Figure 2). Under hypoglycemic conditions, cells with a low energy charge switch off non-vital anabolic processes, i.e., HA synthesis, and concomitantly switch on lipid catabolism via the action of the AMP-activated protein kinase (AMPK), which can increase the ATP/AMP ratio and overcome energetic stress. In particular, AMPK acts on the phosphorylation consensus motif Thr-110 in HAS2. Chemically, AMPK can be controlled by several compounds, such as 2-deoxyglucose, AICAR, and metformin, a medication already used in the clinic for diabetes that causes an inhibition of HAS2 and, in turn, a decrease in HA [54].

The HASes substrate UDP-GlcNAc is also considered a sensor of the metabolic status of the cells [55], as its synthesis depends not only on sugars, but also on amino acids, fatty acids, and nucleotide metabolism (Figure 2). Furthermore, UDP-GlcNAc is necessary for nuclear and cytosolic protein regulation as the donor of the sugar used for post-translational modification of serine and threonine residues, i.e., O-GlcNAcylation made by O-GlcNAc transferase (OGT). This post-translational modification greatly influences the regulation of cellular functions, as various phosphorylation sites are alternatively modified with O-GlcNAc.

Concerning HA metabolism, HAS2 stability is strongly regulated by O-GlcNAcylation but, on the other hand, an efficient HA synthesis can alter the intracellular level of UDP-GlcNAc that, in turn, can affect proteins O-GlcNacylation [56]. In aortic smooth muscle cells (AoSMCs), O-GlcNAcylation upregulates HA synthesis, increasing proliferation via thrombospondin-1 [57] and migration [58].

Although the quantity of UDP-sugar precursors is critical for HA and CS synthesis, their effect at post-translational modification due to AMPK, which means phosphorylation, and O-GlcNacylation, was only found on HAS2, and therefore on HA deposition [38].

Apart from its natural substrates, several factors can alter HAS2 transcription, in particular transcription factors activated by external stimuli, such as TSG-6, Epidermal growth factor (EGF), trans-retinoic acid (RA), PDGF-BB, IL-1, TGFβ, and β-adrenergic agonists, as examples [59,60,61,62]. Moreover, several factors modulating ER stress (e.g.; tunicamycin) or vesicle recycling (e.g.; suramine) can induce intracellular activation of HAS2, which could be related to an increased capability of cellular ECM to interact with immune cells, thus favoring inflammation [63]. During protein maturation, several specific residues have been described as critical for HAS2 regulation; normal posttranslational modifications occurring on these aminoacidic sites not only modulate HAS2 enzymatic activity, but also its progress through the secretory pathway, membrane polymerization, recycling, or even secretion in extracellular vesicles [64]. The ubiquitination of lysine-190 is critical for both HAS2 dimerization and activity [65]. AMPK modifies threonine-110, inhibiting HA synthesis in the case of ATP shortage in cells. On the other hand, the phosphorylation of threonine-228 is necessary for HAS catalytic capability [66], and HAS2 modification with O-GlcNAc stabilizes the presence of active HAS2 in the plasma membrane for more than 5 h [55], with respect to ca. 17 min without modification.

A new frontier of cancer metabolism control is represented by non-coding RNAs (ncRNA) [67], now described as potential drug targets [68], known as rRNAs (Ribosomal RNAs), tRNAs (Transfer-RNAs), snRNAs (small nuclear RNAs), snoRNAs (small nucleolar RNAs), short (miRNAs, siRNAs, piRNAs), and long non-coding RNAs (lncRNAs). However, as the structures and functions of multiple non-coding transcripts have been better characterized, it has become clear that classification based on this grouping is not very useful. In fact, some lncRNAs can act as competing endogenous RNAs (ceRNAs), acting as miRNA sponges, regulating the distribution of miRNAs on their target mRNAs, thereby derepressing miRNA targets and imposing an additional level of post-transcriptional regulation.

At the mRNA level, several microRNAs (miRNAs) are known to affect HAS2 messenger stability such as miRNA7, miRNA23, and Let-7A. Recently, the long non-coding transcript HAS2-AS1, which belongs to the class of antisense transcripts, was shown to influence the chromatin structure around the HAS2 promoter, eventually affecting HAS2 transcription and HA production. Interestingly, the level of HAS2-AS1 is regulated by several factors, such as nuclear factor kappa-light-chain-enhancer of activated B cells (NF-kB) and HIF-1α, which play a pivotal role in both inflammatory and hypoxic responses [69].

Sharing a short complementarity region with HAS2, HAS2-AS1 has been described to form a duplex with HAS2 mRNA, thus preventing its degradation. Moreover, HAS2-AS1 can also compete with endogenous miRNAs, controlling several biological functions simultaneously, including motility, cell cycle, apoptosis, and metabolism [70].

HAS2-AS1 is expressed in normal and tumor cell lines and displays a variety of effects, including inhibition of cell proliferation and upregulation of invasiveness. The effect of this lncRNA is dependent on the cell line and can act by stabilizing or neutralizing HAS2 transcript or sponging endogenous miRNAs in tumor cells. For example, HAS2-AS1 plays an important role in glioblastoma, oral squamous cell carcinomas, and non-small cell lung cancer, and is tightly regulated by several transcription factors including HIF-1α, NF-kB, Sp1, Sp3, high-mobility group AT-hook 2 (Hmga2), signal transducer and activator of transcription 1 (STAT1), and CAMP responsive element binding protein 1 (CREB1). Additionally, Tong et al. reported another nuclear mechanism of action of HAS2-AS1, which can recruit lysine-specific demethylase 1 (LSD1) to the EPHB3 promoter region, resulting in its inhibition [71,72].

In breast cancer, HAS2-AS1 is expressed at higher levels in ER-negative tumors, and among them, those with elevated expression show better prognosis, with an increase in overall survival. Transcriptome analysis of ER-negative breast cancer cell lines showed that high HAS2-AS1 expression is not directly related to HA metabolism; instead, it controls several other pathways (i.e., apoptosis, proliferation, motility, adhesion, and epithelial-to-mesenchymal transition, EMT) through the regulation of miRNAs, such as miR186-3p [70].

ncRNAs have been largely studied in tumors and researchers have evaluated several of them as cancer-specific targets that can be managed by an antisense-based approach to acting at a specific stage in cancer progression. More easily, these ncRNAs have been investigated also as biomarkers for early diagnosis, for example, in lung cancers [73].

In HA research, the epigenetic studies carried out on breast cancer cell lines are not yet related to a specific cancer stage. Once HAS2 mRNA is translated into a protein, it travels in the secretory pathway to reach the plasma membrane. HASes do not exist in the plasma membrane as monomers but as several combinations of dimers: HAS1-HAS2, HAS2-HAS2, and HAS2-HAS3 complexes [74]. Pathological conditions, such as the cholesterol/oxysterol loads in atherosclerosis, induce endoplasmic reticulum (ER) stress which drives HAS2 level alteration that causes high HA deposition in the tunica intima. Unexpectedly, ER stress does not impede HAS2 from reaching the plasma membrane but induces the activation of cholesterol/oxysterols sensors, thus resulting in HAS2 overexpression [75].

The degradation of the HAS2 protein has not yet been fully investigated. Proteins on the plasma membrane can be degraded in endosomes as well as in proteasomes following polyubiquitination. Nevertheless, it has been described that HAS2 can be degraded in vascular ECs via autophagy evoked by nutrient deprivation, mTOR inhibition, or by the pro-autophagic proteoglycan fragments endorepellin and endostatin [36]. These ECM molecules have a critical role in tumorigenesis and angiogenesis via HAS2 turnover [76], and they are currently under investigation for stopping neoangiogenesis [77]; this finding can be a promising target for controlling the HA-derived vessel sprouting within the cancer stroma.

## 4. Cells Crosstalk Affecting HA Synthesis

The amount of HA in the ECM is highly controlled by soluble factors released during cellular changes under physiological and pathological conditions. At the time being, the literature is still lacking an exhaustive investigation into the communication between tumor and normal cells (except for immune cells as reported above). The co-culture of normal fibroblasts and breast cancer cells has been described to increase stromal HA due to HAS2 overexpression in fibroblasts, in response to a stimulus released by the tumor [14]. Indeed, several soluble factors can induce normal cells to produce HA; for instance, platelet-derived growth factor-BB (PDGF-BB) highly increases HA synthesis in human dermal fibroblasts [78] and in mouse cardiomyocytes, which is normally not observed, whereas in NIH 3T3 fibroblasts, PDGF-BB induces a decrease in HA synthesis with a reverse dose-dependence [79]. Hellman et al. [79] reported that the co-culture of cardiomyocytes with fibroblasts increases the amount of HA in the ECM, demonstrating the crosstalk between these cell types, based on soluble factors.

During cancer progression, fibroblasts are usually activated by the same cytokines and transformed into cancer-associated fibroblasts (CAFs), which exhibit different cell proliferation and motility behaviors. The tumor microenvironment is characterized by an inflamed condition, due to both resident and immune cells’ release of a plethora of soluble factors, each able to alter HA production; HAS2 expression could be regulated by interleukin-1β [80], fibroblast growth factor-2 (FGF-2), PDGF, keratinocyte growth factor (KGF), and epidermal growth factor (EGF) [14] in non-cancer cells, while TGF-β exerts pleiotropic effects on both normal and tumor cells through autocrine and paracrine signaling mechanisms [81]. Even though normal fibroblasts of human or mouse origin, when treated with TGF-β, do not increase HA synthesis [78,79], activated CAFs and myofibroblasts release high amounts of HA in the ECM, and are characterized by an upregulated HAS2 [82].

Recently, it was shown that golgins (vesicle-tethering proteins to the Golgi [83]) may also increase HA synthesis. In particular, the protein c10orf118, found in the conditioned media of breast cancer cell lines, was able to induce HAS2 expression in dermal fibroblasts and stimulate HA synthesis [3]. Although it has golgin-like characteristics, the mechanism by which this type of protein affects HA metabolism is still unclear, as golgins have a high molecular weight and are difficult to be secreted. A hypothesis proposed by Caon et al. [14] is that golgins, which are proteins known to act as protein vehicles, may be secreted through exosomes, activating cell signaling in normal cells that induce HA synthesis. The expression of c10orf118 in specimens from breast cancer patients appears to be inversely related to cancer aggressiveness. Higher expression of c10orf118 was significantly associated with better survival in terms of overall survival, relapse-free survival, and distant metastasis-free survival in estrogen receptors-positive patients, suggesting c10orf118 as a possible protective marker. In this study, Caon et al. demonstrated that targeting the protein c10orf118 with a specific antibody reverses the upregulation of HAS2 in fibroblasts and the deposition of HA, identifying the protein as an interesting target for limiting cancer development [14].

Recently, research has focused on the analysis of exosomes, in terms of proteins and genetic material, as carriers of secreted proteins that may affect cell communication and behavior, and consequently, ECM remodeling. The list of possible materials carried by the extracellular vesicles (EVs, the generic term that refers to all the secreted vesicles among which are exosomes), is long and includes RNAs, membrane-bound proteins, membrane lipids, glycoproteins, etc.; and most of them would, of course, affect the ECM. What seems important in HA studies is that this polymer can enhance the EV formation from plasma membrane protrusions; in particular, HASes, which are active only on the plasma membrane, are known to induce the generation of different plasma membrane protrusions and enhance the shedding of EVs [34]. Everything that can alter HASes expression in cancer cells, such as prostate cancer or glioma cells [34], can, as a consequence, cause the release of those vesicles that greatly enhance the crosstalk within the cancerous niche.

## 5. HA Fragmentation

Various mechanisms within the tissue can activate the synthesis and the fragmentation of HA at the same moment, acting on the HA synthases as well as on the hyaluronidases (HYALs), generating new HA from which fragments can be obtained (i.e., oligoHA). Moreover, the fragmentation of HA can also be a chemical reaction, due to oxidation events [84].

OligoHA can affect other cell types, as previously described for neoangiogenesis [84,85]. It is noteworthy that HA and HA fragments can stimulate immune cells to produce chemokines; oligoHA cause the release of Cxcl2 in a TLR2- and TLR4-dependent manner in macrophages, while, by the same pathway, HA oligomers of 12–16 disaccharides stimulate TNF-α production in dendritic cells [86]. In contrast, TNF-α production in macrophages is caused by the activation of CD44 by HMWHA [87].

According to their enzymatic mechanism, HYALs are hyaluronoglucosidases (EC 3.2.1.35), i.e., they hydrolyze the β(1–4) glycoside bond between GlcNAc and GlcUA. Despite their common catalytic activity, they are very different in their localization and regulation [88]. HYALs activity results from the hydrolysis of HA of all sizes into fragments that are as small as tetra-saccharides. Oligosaccharides within the 5–500 kDa range can induce inflammation and angiogenesis. In contrast, HMWHA (1000–2000 kDa) is present in healthy tissues and is anti-angiogenic and immunosuppressive [21].

In Irie et al., transmembrane protein 2 (TMEM2), the only known transmembrane hyaluronidase, has been reported to degrade HA in osteosarcoma, breast, and prostate cancers, thereby remodeling the surrounding microenvironment in a manner that is favorable for their adhesion, migration, and invasion. We could speculate that TMEM2 could be extremely important in cancers with a pericellular coat made of HA, for example in ovarian cancer, where this localization “suggests a pivotal role of the CD44-HA signaling axis for malignant progression in ovarian cancer” [4].

Irie and co-worker performed a culture on HA-coated coverslips relating the enzyme TMEM2 to the formation of the active focal adhesion [89]; this finding explains why the overall survival was significantly shorter in patients with high TMEM2 expression than in those with low TMEM2 expression [90]. Additionally, it could be of further interest to determine the possible effect of the HA fragments produced on the proliferation and migration of these cells.

Like other pathologies involving inflammatory conditions, cancers also exhibit reactive oxidative/oxygen-derived species at the onset and/or later stages of the disease. As an inflammatory milieu, cancers have various possibilities to produce species that can degrade HA; in particular, nitric oxide and superoxide anion radicals [91] are known to cause HA fragmentation. In this case, the overproduction of HMWHA by tissues can be the reaction of normal cells within tissues to buffer the effect of reactive oxygen/nitrogen species (ROS/NOS) and protect cells from ROS-induced apoptosis [92].

## 6. Conclusions

The metabolism of HA in cancers is the result of a complex series of events that implicate multiple cell types, including cancer cells, fibroblasts, endothelial cells, and immune cells. Together, these events can result in an increase in HA around the tumor mass or pericellular coat-circling cells. The origin of this HA could be either cancer cells or fibroblasts. The synthesized HA can then be fragmented into oligoHA, which have their own effect on cells. Each of these steps is strictly cancer-related, which means that potential markers derived from HA metabolism should be evaluated in a tissue-dependent manner.

Since the metabolism of HA is regulated by signals from several HA receptors (e.g.; CD44, RHAMM, LYVE-1) as well as from inflammatory factors (such as TSG-6, EGF, RA, PDGF-BB, IL-1, TGFβ, and β-adrenergic agonists), all those intracellular signals make it difficult to find a univocal extracellular target for controlling its synthesis. Nevertheless, the convergence of all those signals in the regulation of HAS2 highlights the importance of this enzyme and its major epigenetic controller (HAS2-AS1) in cancer development. The isoenzyme HAS3 shows a very important correlation with several cancer types, but the information about its regulation is still insufficient; nevertheless, the possibility that HAS3 could be linked to the release of EVs is a promising new target to explore.

An important occurrence that also needs more attention is the crosstalk between the cells in the cancer niche and, in particular, the response to the molecules addressing HA synthesis, such as golgin C10orf118, or the exchange of materials through vesicles. In fact, the self-sustainment of cancer metabolism can be enlarged from the metabolic re-use of HA as a source of sugars to include the crosstalk with stromal cells as necessary partners in the deposition and/or fragmentation of HA.

## Figures and Tables

**Figure 1 cancers-15-00798-f001:**
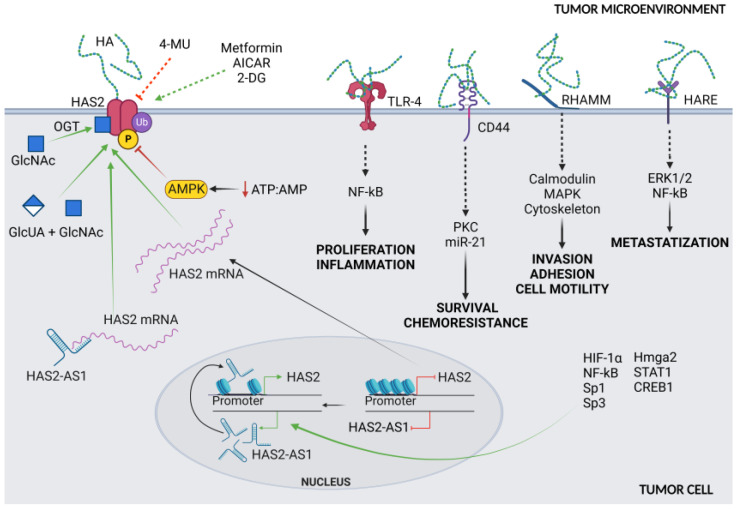
Summary of HA synthesis and its interaction with membrane receptors. The expression of the main synthetic enzyme HAS2 is controlled by various growth and transcription factors, as well as via epigenetic control of transcription by HAS2 antisense 1 (HAS2-AS1). Moreover, the HAS2 activity is also covalently regulated by ubiquitination, phosphorylation, O-GlcNAcylation (due to O-GlcNAc transferase, OGT). Several molecules such as 4-MU, metformin, 2-Deoxy-D-Glucose (2-DG), and the chemical AMPK-stimulating AICAR control HAS2 activity, as demonstrated in in vitro studies.

**Figure 2 cancers-15-00798-f002:**
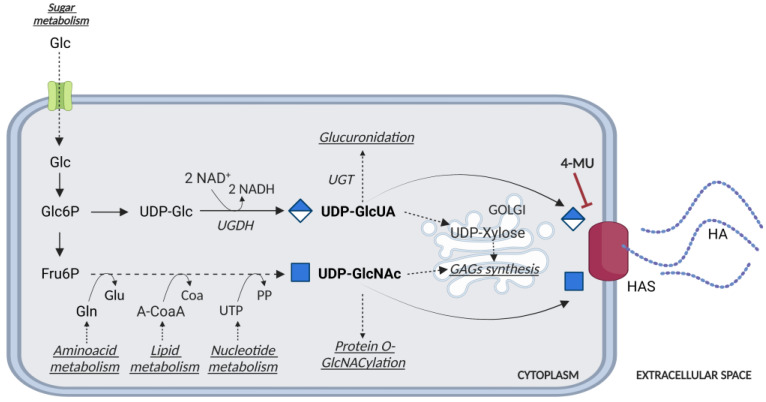
Schematic representation of the UDP-glucuronic acid (GlcUA) and UDP-N-acetylglucosamine (GlcNAc) biosynthetic pathways. Glucose (Glc) enters the cell through the GLUT transporters. Glucose 6-Phosphate (Glc6P) is transformed in UDP-glucose (UDP-Glc) and then in UDP-GlcUA. Fructose 6-phosphate (Fru6P) is directed to the synthesis of UDP-GlcNAc, by a complex metabolic way influenced by amino acids, lipids, and nucleotides availability. UDP-GlcUA and UDP-GlcNAc can be used for the synthesis of HA by HASes enzymes located at the plasma membrane, for glucuronidation and protein O-GlcNAcylation, respectively, or transported inside the ER/Golgi for the synthesis of GAGs and proteoglycans. 4-methylumbelliferone (4-MU) can act as a competitive inhibitor for HASes and decrease the UDP-GlcUA content.

## Data Availability

Figures presented in this review were created with BioRender.com, accessed on 21 November 2022, with agreement numbers CC24O9CRCG, WK24O0CP28, DH24O9FRML to A.P. (Arianna Parnigoni).

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
