# Peer review of "Hyaluronan in the Cancer Cells Microenvironment"

_cancers, 2023, doi:10.3390/cancers15030798_

Round 1
Reviewer 1 Report
To the authors:
This review discusses hyaluronan (HA) synthesis and its regulation, as well as HA in cancer and HA fragmentation. While the sections on HA synthesis and regulation were informative and well-referenced, the sections on HA in cancer and HA fragmentation were more of an overview, lacking specific details and a sharp focus. It almost felt like two separate topics rather than an integrated review, making the goal of the review unclear. Was it the role of HA in cancer, or the regulation of HA synthases, or the regulation of HA synthases in cancer? The summary and abstract could have made this clearer, and the introduction could have better introduced what the review would address.
Overall, I felt the review was a bit disjointed, switching between HA in cancer and HA synthesis and back again. It would benefit from substantial reorganization and revision of some sections to provide a clearer focus and better flow.
Major Comments:
1. Reorganize/change the sections so the flow is better, with each section clearly linked to the next, building on the last. Ensure the goal of the review is the focus and the different sections are integrated.
2. Revise the summary and abstract which should clearly articulate the goal and summary of the review.
3. Revise the introduction to briefly describe the state of the field and what the review will address. Here it would be good to articulate how this review differs from previous reviews from these authors (2017, 2019, 2020).
3.Revise the more general sections (sections 3, 4 and 6) with a sharper focus and more comprehensive analysis, with references as appropriate.
Specific Comments:
1. Consider changing the title –what is meant by flashpoints?
2. In section 3 - what is meant by the sentence line 194-19? Avoid generalized statements with no
references, such as lines 226-230. Instead, inform the reader which cancers/examples you are talking about, and if relevant, identify exceptions/contradictory data.
3. In section 4 – the authors need to check this section carefully for accuracy and expand on their discussion of hyaluronidases, as there are more than just Hyal1 and Ph20. Clarify the localization, specificity and degradation products of Hyal 1 and 2, and mention TMEM2 and CEMIP2 (two newer hyaluronidases).
4. What was the focus of section 6? Consider adding a concluding statement.
5. While the authors are key contributors to this field, there are many self-citations. The authors should ensure that the broader community is referenced where appropriate.
6. There were errors and some sentences did not make sense. The manuscript would benefit from careful editing.
Minor points:
1. Line 30: There are now at least 10 hallmarks of cancer (Hanahan, 2022, Cancer discovery 12, 31-46).
Line 40: 4MU has been used in in vivo experiments, for example (Nagy et al., 2015, J Clin Invest 125, 3928-3940).
2. In Figure 1, does 4-MU decrease UDP-GlcUA concentrations? Should this be indicated on the figure?
3. In Figure 2: what does OGT stand for?
Author Response
We want to thank the reviewer for all the comments. The main concern was that "the review was a bit disjointed, switching between HA in cancer and HA synthesis and back again. It would benefit from substantial reorganization and revision of some sections to provide a clearer focus and better flow"; for answering this point we completely change the sequence of the sections and revised the text in order to avoid duplication of the same statements. For this reason, also the figures are switched.
For what concerns the Specific Comments we modified the title and revised the sections trying to give them a sharper focus; we checked all the references, adding new ones where required and trying to include all the groups working in the HA field. In the hyaluronidase discussion, we added information about TMEM2 that are in line with the focus of the review and of the special issue.
We carefully edit the whole manuscript, and we apologize if there should be other half sentences due to the complete revolution of the text.
Reviewer 2 Report
The paper represents a good overview of the role of HA and HA syntahses in cancer progression
Author Response
thank you for your appreciation. We hope that this second version could also be better for you.
Reviewer 3 Report
While this review contains valuable information, its organization is such that it makes it somewhat difficult to read. The authors should highlight how this adds to what is already known -perhaps offer some thoughts and interpretation of the importance of what is known. It is not clear what is meant by "Flashpoints". Authors should define this which might help clarify the presentation. Several sections are presented as isolated facts and not well integrated.
Author Response
The manuscript has been completely revised, several sections were moved across the text, and repetitions were deleted. The title is changed to make it more transparent and in line with the special issue.
Reviewer 4 Report
Karousou et al reviewed the role hyaluronan, from the glycosaminoglycan family regarding it's role in microenvironment.
1) The title is too board with the use of microenvironment. It would be good to change to cellular or tumor microenvironment.
2) There is a lack of introduction to glycosaminoglycan (GAG) as a whole, then narrowing down to hyaluronan. Please add this in the introduction section. Not all readers are familiar with what is GAG, let alone jumping to hyaluronan directly.
3) There is a need to cite proper references for page 2, line 63-69.
4) There is a need to cite proper references for page 6, line 254-259.
5) Please add new literature to section 5. PMID 32472122; 31847129; 34678263.
6) An author's point of view on how targeting hyaluronan in the tumor microenvironment can be helpful in the future.
7) A conclusion section should be added
Author Response
Thanks to the reviewer for the helpful revision.
The title was changed and we think this version is more focused and in line with the special issue. We added in the introduction background information about glycosaminoglycan (GAG) and concluded the manuscript with a brief conclusion section.
We checked all the references and added the suggested ones and others in the sentences with no explanation.
The text was also revised completely under the suggestion of the other reviewers, so the paragraphs were changed (with the figures) and duplication eliminated.
Round 2
Reviewer 1 Report
The review is substantially improved, with a better flow that is much easier to follow.
However, the English and spelling is not good and the meaning of some sentences is not clear. The manuscript will require extensive editing.
There are a few minor concerns to address below:
Please update Line 69 stating that 4-MU has been “proposed” as an in vivo HA inhibitor. This has been updated in later sections of the manuscript to indicate that it has been used as a HA inhibitor in vivo, and should be changed here for consistency.
HAS 3 produces shorter HA chains than HAS 2. This is not clear from the authors' statement “, HAS2 and HAS3 also differ in the size of HA synthesized, 0.2–2.0 MDa vs >2 MDa [23]. Please clarify.
Line 164 – should this refer to Figure 2, not Figure 1?
Line 186 – An additional reference should be added for the in vitro studies.
Line 397 –"Hyal 2 is located in lysosomes" is not referenced. My understanding is that Hyal1, not Hyal 2, is located in the lysosome. Please clarify and add a reference to support your statement.
Author Response
Reviewer #1
The review is substantially improved, with a better flow that is much easier to follow. However, the English and spelling is not good and the meaning of some sentences is not clear. The manuscript will require extensive editing.
We thank the Reviewer for this comment. We check all the manuscript in order control the English and spelling rewriting the sentences that are not clear.
Please update Line 69 stating that 4-MU has been “proposed” as an in vivo HA inhibitor. This has been updated in later sections of the manuscript to indicate that it has been used as a HA inhibitor in vivo, and should be changed here for consistency.
We thank the Reviewer for highlighting this inconsistency. In this updated version of the manuscript, the sentence is “Hymecromone (i.e., 4-methylumbelliferone, 4-MU), already largely used in in vitro experiments, acting as an in vivo HA inhibitor in diseases characterized by high HA deposition …”
HAS 3 produces shorter HA chains than HAS 2. This is not clear from the authors' statement “, HAS2 and HAS3 also differ in the size of HA synthesized, 0.2–2.0 MDa vs >2 MDa [23]. Please clarify.
We thank the Reviewer for highlighting this not clear sentence. In this updated version of the manuscript, the sentence is “In fact, apart from the differences in the kinetics and regulation, HAS2 and HAS3 also differ in the size of HA synthesized; Has2 produces polysaccharides larger than2 MDa whereas HAS3 synthesizes molecules ranging from 0.2to 2.0 MDa [21].”
Line 164 – should this refer to Figure 2, not Figure 1?
We thank the Reviewer for highlighting this inconsistency. In this updated version of the manuscript, we refer to figure 2.
Line 186 – An additional reference should be added for the in vitro studies.
We thank the Reviewer for highlighting this point. In this updated version of the manuscript, we added proper references.
Line 397 –"Hyal 2 is located in lysosomes" is not referenced. My understanding is that Hyal1, not Hyal 2, is located in the lysosome. Please clarify and add a reference to support your statement.
We thank the Reviewer for highlighting this point. In this updated version of the manuscript, we modified the sentence in a more general way. We also added a comprehensive reference on Hyals (Hyaluronan as an immune regulator in human diseases. Jiang D, et al. Physiol Rev. 2011).

Reviewer 3 Report
While the manuscript has been improved, one is still struck by the feeling that this is just another review of the many that have been written on this topic and it is difficult to understand what is new. The authors could help by addressing what they see as new. Perhaps presenting a "historical " perspective of the discovery of HA in cancer and identify significant areas to explore might add interest.
Author Response
Reviewer #3:
While the manuscript has been improved, one is still struck by the feeling that this is just another review of the many that have been written on this topic and it is difficult to understand what is new. The authors could help by addressing what they see as new. Perhaps presenting a "historical " perspective of the discovery of HA in cancer and identify significant areas to explore might add interest.
We thank the reviewer for all his/her comments and suggestions for improving the readability of the review. In this version of the manuscript we carefully check the text and, as suggested, added some lines mainly in the introduction reporting a “historical” perspective of the HA role in cancer research with the appropriate references (lines 55-69).

Round 3
Reviewer 3 Report
This is now acceptable.